# Driving Behavior Modeling Based on Consistent Variable Selection in a PWARX Model

Jude Chibuike Nwadiuto *[ID], Hiroyuki Okuda[ID] and Tatsuya Suzuki *

Department of Mechanical Systems Engineering Nagoya University, Furocho Chikusa Ward, Nagoya 464-8601, Japan; h_okuda@nuem.nagoya-u.ac.jp
* Correspondence: nwadiuto.jude.chibuike@b.mbox.nagoya-u.ac.jp (J.C.N.); t_suzuki@nuem.nagoya-u.ac.jp (T.S.); Tel.: +81-90-8181-1939 (J.C.N.)

**Abstract:** This paper proposes the hybrid system model identified by a PWARX (piecewise affine autoregressive exogenous) model for modeling human driving behavior. In the proposed model, the mode segmentation is carried out automatically and the optimal number of modes is decided by a novel methodology based on consistent variable selection. In addition, model flexibility is added within the ARX (autoregressive exogenous) partitions in the form of statistical variable selection. The proposed method is able to capture both the decision-making and motion-control facets of the driving behavior. The resulting model is an optimal basal model which is not affected by the choice of data, where the explanatory variables are allowed to vary within each ARX region, thus, allowing a higher-level understanding of the motion-control aspect of the driving behavior, as well as explaining the driver's decision-making. The proposed model is applied to model the car-following driving task based on real-road driving data, as well as to ROS-CARLA-based car-following simulation and compared to Gipp's driver model. Obtained results that show better performance both on prediction performance and mimicking actual real-road driving demonstrates and validates the usefulness of the model.

**Keywords:** driving behavior; driver model; hybrid system identification; statistical variable selection; consistent variable selection; PWARX model



## 1. Introduction

The automotive industry has seen significant advances in autonomous driving and advanced driving assistance systems (ADAS) in recent times [1], not only from a technological point of view but also from a regulatory point of view [2]. These advances have been spurred on by corresponding advances in sensor technology [3]. These sensors such as LiDAR (Light Detection and Ranging), radar, camera, ultrasonic, GPS (Global Positioning System), and IMU (Inertial Navigation System) in combination with sensor fusion have made automated driving possible. Generally speaking, there are six levels of automation in autonomous driving according to the US Society of Automotive Engineers [4]. To the best of our knowledge, as of 2019, level 3 automation has been the maximum level of automation of commercially available vehicles. However, in places like the US, Netherlands, Germany, and the UK, testing of level 6 (full) automation is allowed under certain conditions. The emergence of open-source driving simulators that support flexible specification of sensor suites and environmental conditions such as the CARLA Simulator [5], and LGSVL Simulator [6] have also contributed to the rapid development of autonomous driving. These simulators give full control of the traffic actors, thus making it possible for researchers to easily simulate different traffic scenarios for testing automated solutions. However, these simulators are still not able to fully express the dynamical environment and behavior of real-world traffic. Furthermore, recent advances geared towards decision-making architectures for automated driving [7–9] on a real road means that higher levels (level 4) of automation should be expected in the near future.

Judging from these recent advances in the autonomous driving technology and design of advanced driving assistance systems (ADAS), the natural evolution will be geared towards mixed autonomous driving (i.e., a mixture of human-driven vehicles and automated vehicles), as well as how to reflect personalized driver characteristics and preferences in automated vehicles. Automated vehicles and ADAS need to understand and predict the cognitive and behavioral aspects of the human driver. This means that driving behavior based on real-world driving must be carefully analyzed and modeled. In other words, developing driver models that are more realistic, mathematically rigorous enough, and are capable of expressing the dynamical characteristics of the driving behavior will be the key to meeting this advancement.

From the viewpoint of control technology, numerous ideas have been leveraged to model the human driving behavior [10–19]. In those works, to analyze the driving behavior, the concept of regarding the driver as a sort of controller in combination with linear control theory was adopted. However, a linear controller could be limited in the cases that the driver incorporates not only simple reflexive motion but also decision-making in operating the car. To this end, nonlinear and stochastic modeling techniques like neural networks and hidden Markov models (HMM) were developed for human behavior modeling [20–32]. However, a careful consideration of some applications, such as trying to estimate both the mental and physical states of the driver or interpreting obtained information for ADAS design, reveal the shortcomings of these techniques.

The works [33–37], have shown that, in analyzing the human behavior, the driver timely switches the simple control laws rather than using the complex nonlinear control law. The Hybrid System (HS) is the result of mathematical formulation of this idea, where the transition between modes regarded as the driver's decision-making, are expressed by a discrete-event-driven system and the dynamical characteristics basal to each mode is represented by the continuous-time-driven system. Therefore, through the application of hybrid system modeling, the decision-making and motion-control facets can be simultaneously synthesized, leading to higher-level understanding of human driving behavior.

The piecewise affine autoregressive exogenous (PWARX) model is commonly used as the mathematical model to identify hybrid systems. Numerous algorithms have been proposed for the PWARX model identification [38–43]. However, the issue of estimating or selecting the optimal number of sub-models (modes) is still persistent [44–46].

Considering driving behavior modeling in particular, recent works have demonstrated the effectiveness of using mode segmentation [47–49]. In [47], a hierarchical PWARX model was used to model driving behavior, [48] exploited the PWARX model for the design of an assist system for the driver and [49] used a stochastic ARX (SS-ARX) to model the human driving behavior. The assumption in these works is that the explanatory variables are consistent across all modes (i.e., the explanatory variables do not change from one ARX region to another). The drawback of this assumption is that the flexibility of the model is reduced, hence increasing the likelihood of the model performance being restricted. In addition, if the explanatory variables are allowed to vary from one mode to another, by introducing, for example, statistical variable selection [50], the motion-control facet could be analyzed and understood at a higher level. Therefore, the resulting model is simpler and facilitates online application because of the reduced computational cost. In addition, personalized driver preferences could easily be identified and embedded for ADAS design. The results shown in [47–49] are questionable to apply in the real world since driving simulator data, instead of real-road driving data, were used in these works. Lastly, the issue of how to decide the optimal number of modes was not addressed, and the number of modes was chosen manually.

In [51], Nwadiuto et al. demonstrated the usefulness of allowing flexibility in form of statistical variable selection within the hybrid system modeling when modeling driving behavior. In particular, driving behavior in the motion facets could be understood at a higher level with results showing that the variables influencing the dynamics in each region change from one region to another. However, the mode segmentation was carried out

manually based on the range band. This gives little insight into understanding the driver's decision making and model structure (transition between modes) in general. In addition, the robustness of the model to change in data was not considered. Giving that driving behavior changes over time depending on surrounding environment, driving time, driver's mood, etc., developing a model which is consistent in the sense that the model should be robust to a slight change in the training data should be addressed for automated driving and designing driver's assistance systems.

In this paper, a hybrid system identified by a PWARX model for modeling human driving behavior is proposed, wherein the mode segmentation is carried out automatically, and the optimal number of modes is decided by a novel methodology based on consistent variable selection (Section 2.4). In addition, model flexibility was added within the ARX partitions in the form of statistical variable selection. The resulting model is an optimal basal model that is not affected by a slight change in the training data, where the explanatory variables are allowed to vary within each ARX region.

In the proposed model, the driving data used were collected on a real road which is one of the important points of this work. Information that expresses the relationship between the ego vehicle and surrounding traffic agents such as distance to the leading vehicle, time to collision (TTC), range rate, etc. are collected using 3D LiDAR and camera sensors. Additionally, the information relating only to the ego vehicle such as vehicle speed, braking and accelerator pedal operation, steering wheel angle is collected using an OBD-CAN reader connected to the vehicle. These information capture how the driver behaves in relation to the surrounding traffic agents, and thus are considered as parameters for the model. The details of the driving experiment and the model input and output variables are described fully in Sections 3.1 and 3.2, respectively. First, feature vector extraction is introduced in order to classify the data based on the similarities of the underlying dynamics, then weighted k-means was used for clustering. The optimal number of modes was decided by a new method that considers the consistency in the variable selection within each mode. Variable selection is carried out using the Bayesian Information Criterion (BIC), while the parameter estimation and the separating hyper-planes identification are achieved using the least squares method and the support vector machines (SVMs), respectively. Finally, the usefulness of the proposed model is demonstrated through real-road driving data analysis and modeling, and verified by simulation application using a combination of a ROS (Robotic Operating System) and CARLA driving simulator framework.

## 2. Methodology

### 2.1. Overview

The framework presented here leverages the hypothesis that the human driver appropriately switches the simple control laws, which is realised by grouping together the areas of common dynamical characteristics into individual sub-models and identifying the conditions governing the transition between them. In the proposed methodology, the PWARX (piecewise affine autoregressive exogenous) model was used to capture the dynamical characteristics of each sub-model. To understand the human driver better, as well as to reflect personalized driver facet, flexibility in the form of variable selection was introduced within each sub-model. For deciding the appropriate number of sub-models, a novel method based on the consistency of variable selection, which is one of the main contributions of this paper, was introduced. Finally, the support vector machine was used to identity the hyperplanes separating the sub-models. The overview of the framework presented in this study is shown in Figure 1, which consists of the system identification process and how the identified model was evaluated.

The proposed modeling framework is applicable to different human driving tasks, therefore only the system identification process is discussed in this section without a focus on application to any specific driving task. In this paper, the proposed model is tested on the car-following driving task on a downtown road. The boundary conditions, model

assumptions, hyper-parameters, input and output variables for the car-following driving task will be discussed in Section 3.

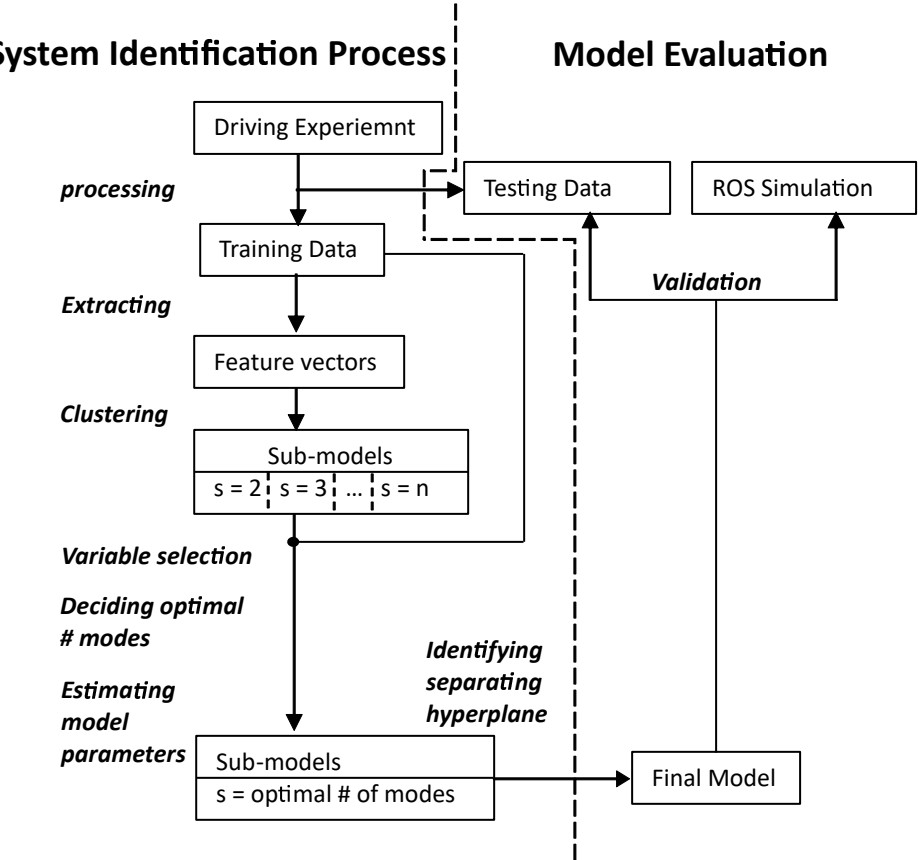

**Figure 1.** Overview of the proposed Framework.

### 2.2. PWARX Model

The PWARX (piecewise affine autoregressive exogenous) model is widely used as a mathematical model for identifying hybrid systems and is given by

$$
y_k = \begin{cases} \theta_1^\mathsf{T} \begin{bmatrix} r_k \\ 1 \end{bmatrix} + e_k, & \text{if } r_k \in \mathscr{X}_1 \\ \vdots & \vdots \\ \theta_s^\mathsf{T} \begin{bmatrix} r_k \\ 1 \end{bmatrix} + e_k, & \text{if } r_k \in \mathscr{X}_s \end{cases}
\tag{1}
$$

for $k = 1, 2, \ldots, N$. The input, output, and noise at time $k$ are given by $u_k \in \mathbb{R}^m$, $y_k \in \mathbb{R}^p$ and $e_k \in \mathbb{R}^p$, respectively, whereas the regressor vector $r_k \in \mathbb{R}^n$ is given by $r_k = [y_{k-1}^\mathsf{T} \, y_{k-2}^\mathsf{T} \, \cdots \, y_{k-n_y}^\mathsf{T} \, u_{k-1}^\mathsf{T} \, u_{k-2}^\mathsf{T} \, \cdots \, u_{k-n_u}^\mathsf{T}]^\mathsf{T}$, where $n = pn_y + mn_u$, and $m$, $p$, $n$ are the dimensions of the real coordinate space of $u_k$, $y_k$, $r_k$, respectively, while $n_y$, $n_u$ are non-negative integers. $\mathscr{X}_i$ for $i = 1, 2, \ldots, s$ denotes the convex polyhedral subsets of the regression space $\mathscr{X} \subseteq \mathbb{R}^n$. $s$ represents the number of sub-models and $\theta_i \in \mathbb{R}^{(n+1) \times p}$ for $i = 1, 2, \ldots, s$ are the model parameters to be estimated.

It should be noted that the PWARX model used in this work is a modified version of the general PWARX model presented above, in the sense that flexibility is allowed by variable selection in the number of variables from one sub-model to another; as such, the number of estimated parameters $\theta_i \in \mathbb{R}^{(n+1) \times p}$ in each sub-model can be different.

### 2.3. Data Clustering

#### 2.3.1. Feature Vector Extraction

The main purpose of introducing the feature extraction here is to try to classify the data based not only on the data value but also on the similarity of the underlying dynamics. This thinking is formally supported by the assumption that if a PWA map is locally linear, then a small subset of points $u_k$ which are close to each other will likely be in a similar region. Therefore, at first, a transformation from the data space to the feature vector space based on the local dynamics of the data is carried out. This transformation from the data space to the feature space further separates the data, thereby making it easy for the data to be clustered, thus highlighting the reason why feature vector extraction was performed on the data. It should be noted that the variable names or definitions here are particular to only this sub-section. The procedure for this transformation is as follows:

Given $(y(j), x(j))$ a set of sample data for $j = 1, 2, \ldots, n$, the neighboring $c$ data points in the $(y, x)$ space, where $c$ is a positive integer less than $n - 1$, is collected for each $(y(j), x(j))$ to generate a local data set $LDs_j$ and calculate the feature vector $\xi_j$, where the index $j$ denotes the order in the data space. The feature vector is given by

$$\xi_j = ((\theta_{j,1}^{LD})^\mathsf{T}, (\theta_{j,2}^{LD})^\mathsf{T}, m_j^\mathsf{T})^\mathsf{T} \tag{2}$$

where $((\theta_{j,1}^{LD})^\mathsf{T}, (\theta_{j,2}^{LD})^\mathsf{T})^\mathsf{T}$ and $m_j$ are the local parameters in the local ARX model and the mean value of the data $u$, respectively, in the $LDs_j$, and are calculated as follows:

$$\theta_{j,l}^{LD} = (\phi_j^\mathsf{T} \phi_j)^{-1} \phi_j^\mathsf{T} y_{LDs_j,l}, \tag{3}$$

$$m_j = \frac{1}{c} \sum_{u \in LDs_j} u, \tag{4}$$

where $y_{LDs_j,l}(c \times 1; l = 1, 2)$ is the $LDs_j$ output samples, and $\phi_j$ is denoted by

$$\phi_j = (u_1\, u_2 \ldots u_c)^\mathsf{T}, \text{for } u \in LDs_j. \tag{5}$$

Finally, for each feature vector $\xi$, a covariance matrix $R_j$, which is used as a weighting matrix for computing the dissimilarity between feature vectors used in the clustering procedure, is computed as:

$$R_j = \begin{bmatrix} V_{j,1} & 0 & 0 \\ 0 & V_{j,2} & 0 \\ 0 & 0 & Q_j \end{bmatrix} \tag{6}$$

where

$$V_{j,l} = \frac{SSR_{j,l}}{c - (9 + 1)} (\phi_j^\mathsf{T} \phi_j)^{-1} \tag{7}$$

$$SSR_{j,l} = y_{LDs_j,l}^\mathsf{T} (I - \phi_j (\phi_j^\mathsf{T} \phi_j)^{-1} \phi_j^\mathsf{T}) y_{LDs_j,l} \tag{8}$$

$$Q_j = \sum_{u \in LDs_j} (u - m_j)(u - m_j)^\mathsf{T}. \tag{9}$$

#### 2.3.2. Clustering Algorithm

In this paper, the weighted k-means clustering was implemented for clustering the data. The weights calculated during the feature vector extraction as given in Section 2.3.1, which improves robustness to outliers and the fast convergence of the k-means algorithm were reasons this method of clustering was chosen. Generally speaking, any kind of clustering algorithm could be used.

### 2.4. Optimal Number of Sub-Models

For deciding the optimal number of sub-models or modes, a novel method that uses the consistency of the variable selection result is proposed. The hypothesis here is that the true model for a driver should be replicative; thus, in this study, optimality is defined as the most structural–consistent identified hybrid system model, which in fact can be interpreted as the optimum trade-off between the model robustness and model error. The step-by-step procedure of the novel method is given in Algorithm 1.

The most important detail of Algorithm 1 can be summarized as, taking a cluster, randomly divide or partition the cluster into some number of folds. For each fold, variable selection using BIC is carried out, and the results of the variable selection are compared between the folds. The best cluster is the cluster that replicates the same variable selection results amongst the folds. A pictorial example of using consistent variable selection for deciding the optimal number of sub-models or modes is shown in Figure 2.

---

**Algorithm 1** Deciding the optimum number of sub-models

---

**Input:** $K - 1$ sets of clusters, $C_{i=1,\ldots,s}$ for $s = 2, \ldots, K$
**Output:** Optimal number of sub-models, $opt_s$
    *Initialisation* :
  1: Given the Input
  2: **for** $p = 1$ to $P$-number of times **do**
  3:   **for** $s = 2$ to $K$ **do**
  4:     **for** $i = 1$ to $s$ **do**
  5:       Randomly partition $C_i$ into $n$ folds
  6:       **for** $q = 1$ to $n$ **do**
  7:         Perform variable selection using BIC on $C_i^q$
  8:         Store variable selection result in $V_i^q$
  9:       **end for**
10:       Let $V_i^{q=1,\ldots,n} = V_i^{r=1,\ldots,n}$; and $T_i^{q,r} = 1$ if $V_i^q = V_i^r$, or 0 otherwise
11:       Compute $freq_{C_i} = \dfrac{1}{n^2} \sum_{r=1}^{n} \sum_{q=1}^{n} T_i^{q,r}$
12:     **end for**
13:     Compute $avg.freq_s = \dfrac{1}{s} \sum_{i=1}^{s} freq_{C_i}$
14:   **end for**
15:   Store $temp_{opt_s}^p = \max_s (ave.freq_s)$
16: **end for**
17: **return** $opt_s = \text{mode}(temp_{opt_s}^{p=1,\ldots,P})$

---

### 2.5. Variable Selection and Identification of Parameters and Hyperplanes

Generally speaking in data analysis, the ways of choosing data for a model depending on the application can be either by intuitive selection, by using every measured datum, or by choosing the essential data. In this paper, the essential data are chosen as input through variable selection in order to reduce the complexity of the model while keeping the accuracy. Several methods have been introduced for the variable selection. The information-based criteria such as the Akaike information criterion (AIC), the variant of AIC, AICc, and the Bayesian information criterion (BIC) are widely used in statistical model selection [50]. In this work, the BIC was adopted because it penalizes model complexity the most amongst the information criteria-based method for variable selection. The goal here is to achieve the model that is consistent and basal (basic) to the driver, thus the choice of BIC. Generally speaking, BIC establishes the relationship between the relative expected Kullback–Leibler distance and the maximum log-likelihood estimation. For regression analysis, BIC is given as

$$\text{BIC} = N.\log\left(\frac{RSS}{N}\right) + K.\log(N) \tag{10}$$

where *RSS* is the residual sum of squares, *N* is the number of observations, *K* is the number of regression parameters and $\log()$ is the natural logarithm. Intuitively, BIC takes a balance between the model accuracy (the first term of Equation (10)) and model complexity (the second term of Equation (10)). The model with the lowest criterion value is chosen as the best model.

The parameter estimation of each sub-model is achieved by the least squares method. Finally, the separating hyperplanes between the sub-models are identified using multi-class support vector machines (SVMs) as described in [52,53].

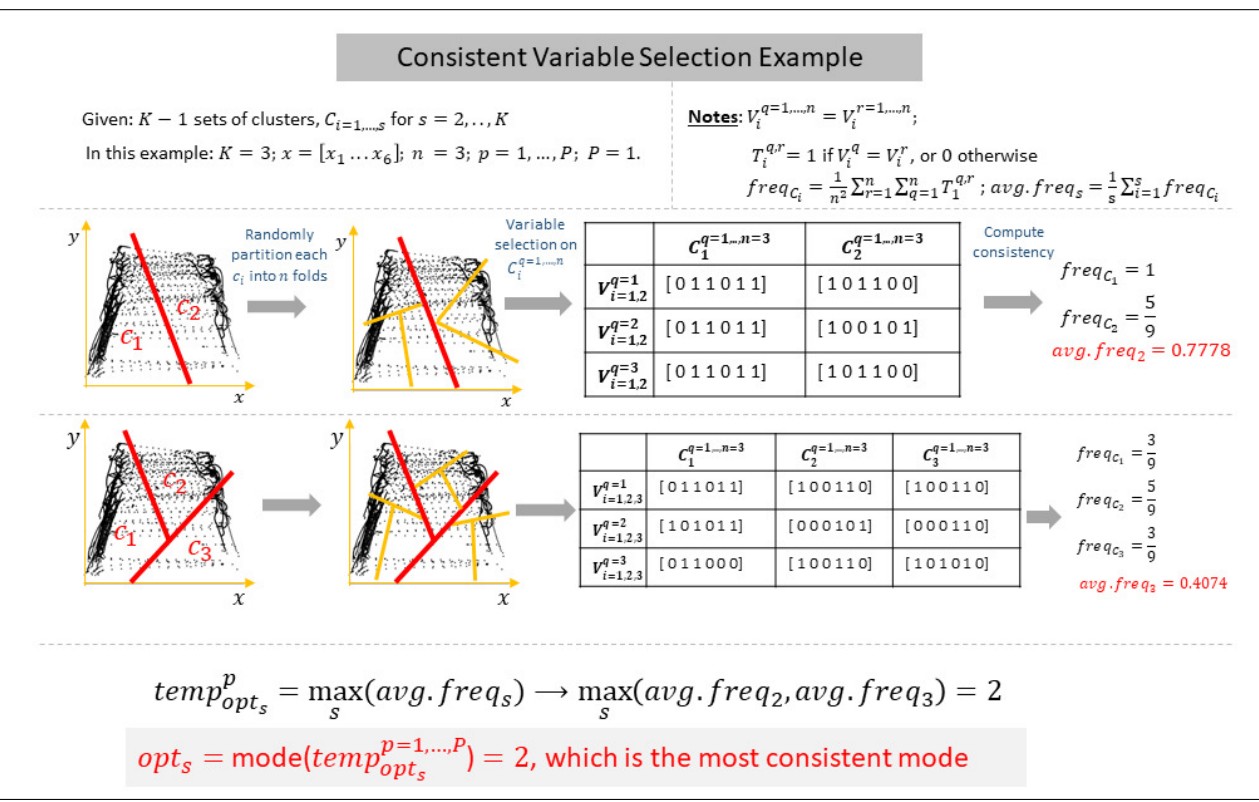

**Figure 2.** An example of selecting the optimal number of modes or sub-models using consistent variable selection, given 2 sets of clusters.

## 3. Analysis and Modeling of Driving Behavior

Human driving behavior can be regarded to be made up of two facets, the first being the decision-making facet characterized by logical switching, and motion-control described by continuous dynamics as the second facet. Thus, it is expected to simultaneously extract these two facets when hybrid system identification is applied to the behavioral driving data. Furthermore, by allowing flexibility in form of variable selection within the identified motion-control facet, logic can be embedded, therefore allowing for further coherency in understanding and interpreting the continuous dynamics. The usefulness of the proposed modeling framework is demonstrated by application to real-road driving behavioral data.

### 3.1. Acquisition of Driving Data

The strong point of the work presented here is the use of data acquired from a real road driving experiment. Most of the works geared towards driving behavior analysis use driving data from a driving simulator because of the relative ease at which experimental data can be collected. However, there is a gap between the driving data from the driving simulator and that from real road driving. This work aims to bridge this gap by developing a model suitable for real road behavioral data; therefore, a real-world driving experiment was designed.

This paper focuses on the driver's car-following task in the downtown road and, accordingly, the location was chosen to be Sakae (GPS coordinates: 35.1698544, 136.9081513) the downtown area or city center of Nagoya city, Japan. This is a typical downtown area with high traffic volume, and with relatively flat topography. The driving route is shown in Figure 3.

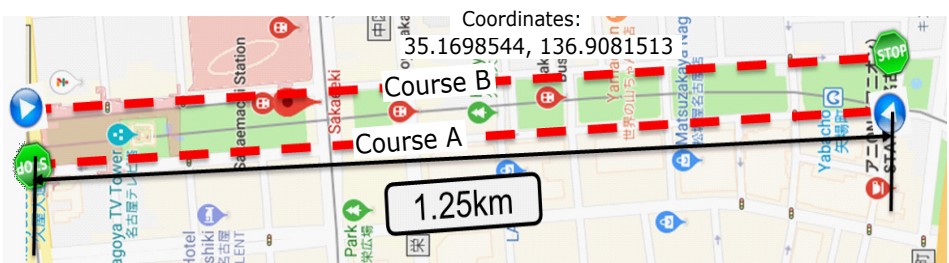

**Figure 3.** Route of the driving experiment (Sakae) showing the two courses (Course A and Course B), with the relative GPS coordinates of the area.

For collecting the driving data, a sensor setup consisting of a 3D LiDAR, a USB camera, and an OBD CAN reader mounted on a Toyota Prius 2015 model was used. Through the OBD CAN reading, driving information such as vehicle speed, throttle pedal position, brake pedal position, and wheel speed was measured. The USB camera was fused with the 3D LiDAR to detect and measure the position of the surrounding traffic entities like the leading vehicle. Figure 4 shows the sensors that were mounted on the car used in this driving experiment.

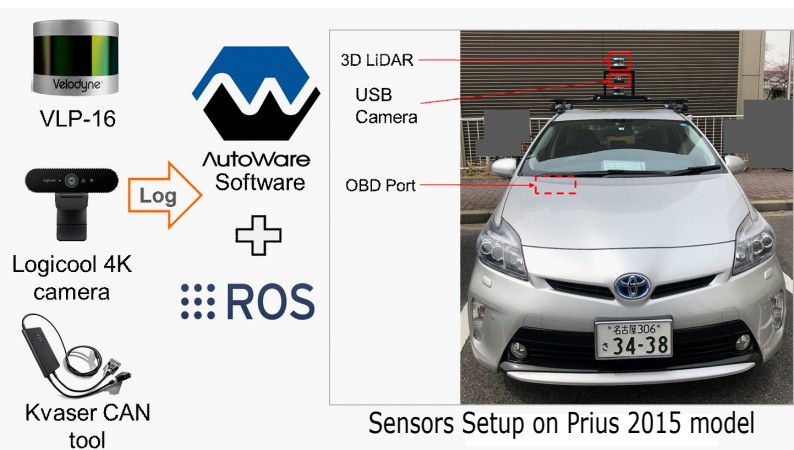

**Figure 4.** Sensor setup for collecting the real-road driving data.

A total of 6 drivers (A, B, C, D, E, and F), each with more than three years of driving experience, took part in this driving experiment. As shown in Figure 3, two courses (A and B) each with a length of 1.3 km were chosen for the driving experiment, with driving conducted at varying times of the day to reflect the change in traffic volume of the selected driving courses. Deliberate consideration was made to ensure that the driving experiment included peak traffic hours. Each driver drove five times on each course totaling about an hour of driving and was instructed to drive normally as they would drive in an everyday situation. Furthermore, since the point of interest was observing car-following driving behavior, the drivers were instructed to keep the middle lane and were not allowed to change lanes throughout the driving experiment. Only the ego (following) vehicle was controlled during the experiment, therefore ensuring a naturalistic interaction between the surrounding entities. This experiment was conducted according to the guidelines of the Declaration of Helsinki, and approved by the Ethical Committee of Graduate School of Engineering, Nagoya University with approval number of 18-6. The video of a sample

run of the driving experiment which shows the actual traffic and environmental conditions is shown https://youtu.be/LBM3PTlfecQ (accessed on 18 May 2021) for course A and https://youtu.be/AeOn3bxFzOQ (accessed on 18 May 2021) for course B.

### 3.2. Definition of Input Candidates and Output

This subsection presents the input candidates and output considered in the model for analyzing and modeling the car-following driving task. It should be noted that the variable names (definitions) are exclusive to this subsection. The driver's candidates for input variables and output variables based on Figure 5 are defined as follows:

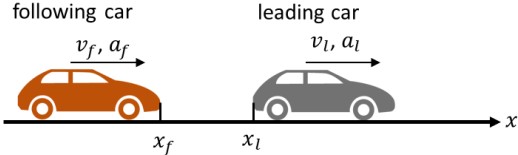

**Figure 5.** Definition of variables between ego car and leading car.

Candidates for input variables

- Range (m): $u_1 = x_l - x_f$;
- Range rate (m/s): $u_2 = v_l - v_f$;
- KdB: $u_3$;
- Jerk (m/s$^3$): $u_4 = \Delta a_f / \Delta t$
- Inverse of time to collision (rate of increase in visual angle to leading car) (s$^{-1}$):
$$u_5 = \frac{v_l - v_f}{x_l - x_f};$$
- Time headway (time deference between cars) (s): $u_6 = \dfrac{x_l - x_f}{v_f}$.

Output variable

- Speed (m/s): $y = v_f$.

KdB is a risk-feeling index which is defined as the logarithm of the time derivative of the leading car's back area projected onto the ego driver's retina [54]. To express KdB, $u_1$ and $u_2$ are used as follows:

$$u_3 = \begin{cases} 10 \times \log(-c), & \text{if } c < -1 \\ -10 \times \log(-c), & \text{if } c > 1 \\ 0, & \text{if } -1 \geq c \leq 1 \end{cases} \tag{11}$$

where $c$ is denoted by

$$c = 4 \times 10^7 \times \frac{u_2}{u_1{}^2}. \tag{12}$$

An intuitive interpretation of KdB is that the level of dangerous situation the driver faces increases as the value of KdB increases. Before identification, all variables were normalized.

### 3.3. Hyper-Parameters Used in the System Identification Process

The hyper-parameters of the data pipeline, as well as the parametric assumptions used in the system identification process for modeling the car-following driving task, are given as follows:

- PWARX model: Input data are range, range rate, KdB, jerk, inverse of time collision, and time headway ($u_1$, $u_2$, $u_3$, $u_4$, $u_5$, and $u_6$). The Model output $y$ is speed, and the number of data points $N$ is approximately 4200 with a time step of 0.1 s for each driver. The first-order dynamics is considered as the controller model; thus, the regressor vector $r_k$ is $[y_{k-1} \, u_{1,k-1} \, u_{2,k-1} \, u_{3,k-1} \, u_{4,k-1} \, u_{5,k-1} \, u_{6,k-1}]^\mathsf{T}$.

- Feature vector extraction: The constant $c$ (number of neighboring data points) is chosen as 200. This value was chosen by trial and error based on the data. As a general rule, 10% of the number of data points, $N$, is a good number for this constant.
- Optimal Number of Sub-models: $K$, $P$ and $n$ (folds) are chosen to be 10, 100 and 3, respectively. These values were chosen intuitively mainly based on the number of data points.

## 4. Modeling Results

### 4.1. Mode Segmentation

The result for the estimated optimal number of modes or sub-models for drivers A–F based on consistent variable selection is shown in Table 1.

**Table 1.** Identified number of modes from consistent variable selection for each driver.

| Driver | A | B | C | D | E | F |
|---|---|---|---|---|---|---|
| Optimal Number of Modes | 3 | 3 | 4 | 3 | 3 | 3 |

Generally speaking, a three-mode model was identified as the optimal number of modes amongst the drivers for the car-following task in the downtown area. The mode segmentation result in the range-KdB space, range rate–range space, and acceleration-KdB are, respectively, shown in Figures 6–8 for drivers A, and B. These figures show the decision/result of the optimal number of modes or sub-models where a unique color represents a unique mode or sub-model with its own distinctive dynamical characteristics. Additionally, the boundaries of where the colors change can be regarded as the mode-switching or decision-making of the driver. In these figures, the red, green, and blue markers correspond to mode (sub-model) 1, 2, and 3, respectively, of Tables 2 and 3. In interpreting this result, mode 1, the region marked in red is characterized by high KdB, negative range rate, as well as the braking operation being activated numerous times, implying that this is the dangerous region for the driver. In mode 2, the green marked region, the KdB is negatively large, the range rate is positive, and the driver mainly operates the accelerator pedal, indicating that this is the safest region for the driver. The area marked in blue, mode 3 is characterized by both positive and negative KdB values, acceleration of this area is near zero with change fairly constant, which implies that this is the cruising region. In this mode, the driver is comfortable and tries to maintain stable dynamics, suggesting that this is the part of the vehicle-following task where the driver is least attentive.

Generally speaking, the car-following driving task in the downtown area can be said to consist of three modes or sub-models, (i.e., the dangerous region, safe region and cruising region. However, the separating hyper-planes (decision boundaries) are different from one driver to another as shown in Figures 6–8. In these figures, parts (a) and (b) represent the identified modes for driver A and driver B, respectively. In comparing drivers A and B, driver A can be said to be a more conservative driver in the sense that the area of the identified dangerous region is bigger. In designing automated driving systems, for example for drivers A and B, this unique characteristic particular to each driver can be leveraged to offer a more comfortable driving experience.

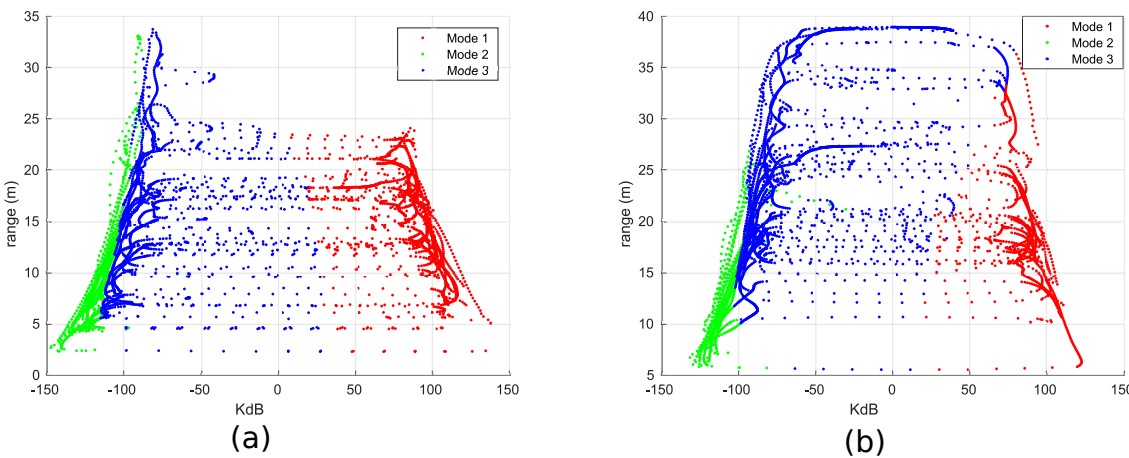

**Figure 6.** Mode segmentation (optimal number of sub-models) result of the car-following task in the range-KdB space, where a unique color represents a unique mode (sub-model) with its own distinctive motion-control characteristics: (**a**) Three-mode model identified for driver A; (**b**) three-mode model identified for driver B.

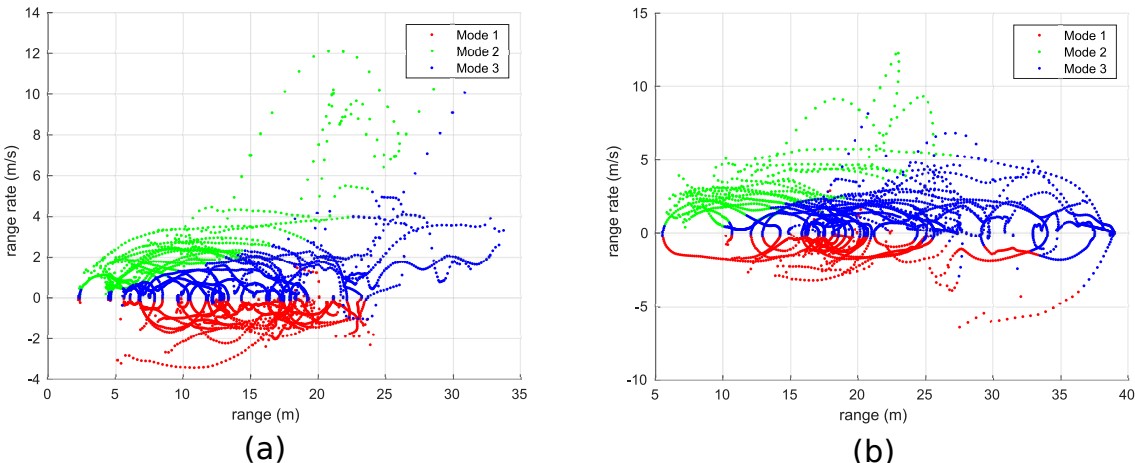

**Figure 7.** Mode segmentation result (optimal number of sub-models) of the car-following task in the range rate–range space, where a unique color represents a unique mode (sub-model) with its own distinctive motion-control characteristics: (**a**) three-mode model identified for driver A; (**b**) three-mode model identified for driver B.

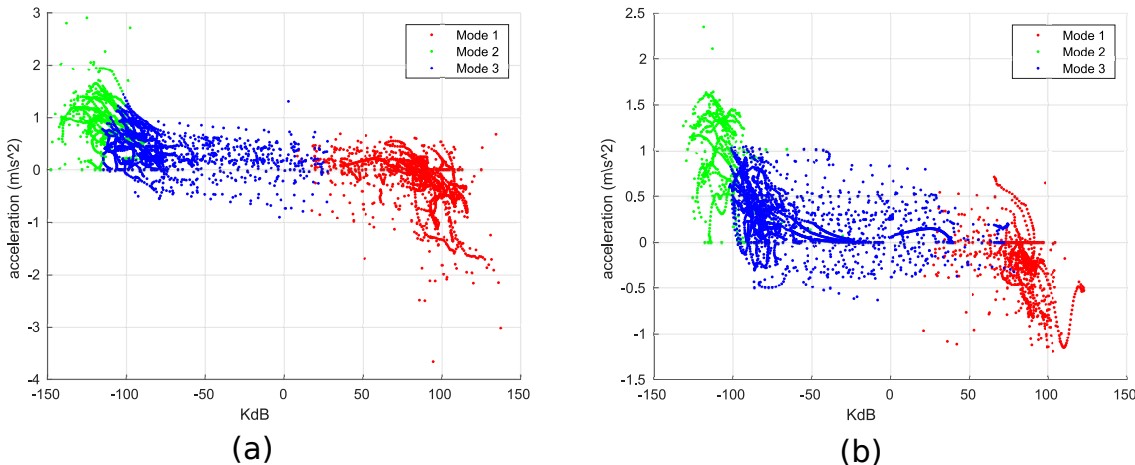

**Figure 8.** Mode segmentation result (optimal number of sub-models) of the car-following task in the acceleration-KdB space, where a unique color represents a unique mode (sub-model) with it's own distinctive motion-control characteristics: (**a**) three-mode model identified for driver A; (**b**) three-mode model identified for driver B.

The mean values of the variables in each mode or sub-model for all the drivers are shown in Table 2, whereas Table 3 shows the acceleration statistics for each mode. These tables show the location of each mode, therefore explaining the physical meaning of each mode. Mode 1 is in the high positive KdB, negative range rate and largely negative-acceleration (deceleration) region. On the other hand, mode 2 is at the opposite spectrum of mode 1, i.e., high negative KdB, positive range rate, and positive acceleration, whereas mode 3 is located where the KdB, range rate and acceleration are distributed uniformly between positive and negative sides. The vehicle-following task in the downtown area can be said to be a cyclic of safe-departure by the leading vehicle, cruising at desirable dynamics by the ego car and cautious approach to the leading car. Furthermore, examining the figures of the segmentation and the statistic tables, it can be concluded that amongst the input variables for modeling the vehicle-following task, KdB ($u_3$) and range rate ($u_2$) strongly affect the decision making (mode segmentation) of the driver, whereas the other variables influence the motion dynamics. Examining these mean values in Table 2 for drivers A to F, it could be seen that the values are different for all the modes from driver to driver. In addition, the variance values in Table 3 show the variation in the way the driver operates the brake and accelerator pedals. These variations (pedal behavior) in each node are also different from driver to driver. This signifies that each driver has his own unique model structure (i.e., unique decision making) for the car-following task, and thus this uniqueness can be leveraged directly to design automated systems peculiar to each driver.

**Table 2.** Mean values of the variables in each mode $i$.

| Driver | Mode ($i$) | $u_1$ | $u_2$ | $u_3$ | $u_4$ | $u_5$ | $u_6$ | $y$ |
|---|---|---|---|---|---|---|---|---|
| | 1 | 14.7967 | −0.7365 | 82.9239 | −0.0947 | 0.0573 | 1.6321 | 9.0802 |
| A | 2 | 10.1805 | 2.5240 | −115.3931 | −0.1916 | −0.2338 | 2.4917 | 4.5813 |
| | 3 | 15.2099 | 0.9024 | −76.5456 | −0.0726 | −0.0564 | 1.7508 | 8.8569 |
| | 1 | 19.1161 | −1.0761 | 82.1025 | −0.1442 | 0.0599 | 2.8145 | 6.8006 |
| B | 2 | 13.2032 | 3.0255 | −107.8165 | −0.0773 | −0.2275 | 5.1354 | 3.2444 |
| | 3 | 25.3100 | 0.9317 | −55.9455 | −0.0544 | −0.0425 | 3.5853 | 7.1059 |
| | 1 | 22.5530 | −1.0362 | 75.2793 | −0.1437 | 0.0507 | 3.1521 | 7.2707 |
| C | 2 | 12.4296 | 3.0010 | −111.8950 | 0.2716 | −0.2467 | 6.1057 | 2.6634 |
| | 3 | 26.6439 | 1.7252 | −76.8184 | −0.0072 | −0.0784 | 4.7373 | 6.0309 |
| | 4 | 22.5461 | 0.7366 | −65.0536 | −0.0227 | −0.0341 | 2.9458 | 7.8383 |
| | 1 | 7.4822 | 1.4649 | −113.5721 | −0.1556 | −0.1883 | 3.9329 | 2.5935 |
| D | 2 | 10.4366 | −0.6515 | 65.4002 | 0.0329 | 0.0567 | 2.4888 | 4.5814 |
| | 3 | 19.3710 | 0.7052 | −56.4716 | 0.0466 | −0.0444 | 2.6940 | 7.4533 |
| | 1 | 17.9504 | −1.0276 | 80.6962 | −0.2128 | 0.0603 | 2.3383 | 7.7231 |
| E | 2 | 10.6959 | 2.2784 | −112.2794 | −0.3386 | −0.2163 | 3.3595 | 3.8392 |
| | 3 | 23.4378 | 0.8599 | −52.0313 | −0.0258 | −0.0424 | 2.9062 | 8.4229 |
| | 1 | 14.5640 | −0.7783 | 80.2988 | −0.1284 | 0.0632 | 1.6611 | 8.6957 |
| F | 2 | 13.7065 | 1.2122 | −95.2759 | −0.0027 | −0.1033 | 2.0076 | 7.6085 |
| | 3 | 18.1030 | 0.2739 | −25.9114 | −0.1185 | −0.0134 | 1.8595 | 9.6969 |

**Table 3.** Acceleration statistics for each mode $i$.

| Driver | Mode ($i$) | Mean | Median | Min | Max | Variance |
|---|---|---|---|---|---|---|
| | 1 | −0.2138 | −0.0740 | −3.6593 | 0.6906 | 0.2388 |
| A | 2 | 0.9392 | 0.9349 | 0 | 2.9056 | 0.1665 |
| | 3 | 0.3748 | 0.3834 | −0.9020 | 1.4440 | 0.1115 |
| | 1 | −0.2615 | −0.2228 | −1.1884 | 0.7133 | 0.1079 |
| B | 2 | 0.9344 | 0.9809 | 0 | 2.3506 | 0.1787 |
| | 3 | 0.2500 | 0.1964 | −0.6297 | 1.0465 | 0.0890 |

**Table 3.** *Cont.*

| Driver | Mode ($i$) | Mean | Median | Min | Max | Variance |
|---|---|---|---|---|---|---|
| C | 1 | −0.1870 | −0.2072 | −1.0996 | 0.6637 | 0.1017 |
|  | 2 | 0.8761 | 0.8607 | 0.2885 | 1.4045 | 0.0354 |
|  | 3 | 0.4271 | 0.4203 | −0.4586 | 1.0572 | 0.0884 |
|  | 4 | 0.2023 | 0.1794 | −0.6098 | 0.9715 | 0.0752 |
| D | 1 | 1.0527 | −0.2552 | 0 | 0 | −0.0445 |
|  | 2 | 1.2829 | 0.1571 | 0 | 0 | 0 |
|  | 3 | 0.9368 | 0.1388 | 0.0282 | 0 | 0.2396 |
| E | 1 | −0.2177 | −0.2102 | −1.1146 | 0.5746 | 0.0899 |
|  | 2 | 0.7833 | 0.8386 | −0.1192 | 1.7035 | 0.1054 |
|  | 3 | 0.2525 | 0.2473 | −0.7444 | 1.2167 | 0.1151 |
| F | 1 | −0.3308 | −0.2750 | −1.5212 | 0.9163 | 0.1819 |
|  | 2 | 0.6636 | 0.6305 | −0.8367 | 2.0976 | 0.2561 |
|  | 3 | 0.1177 | 0.0589 | −0.6595 | 1.0675 | 0.1290 |

### 4.2. Variable Selection

Statistical variable selection is usually carried out to identify from a list or set of variables the most influential variable or variables; therefore, variable selection was performed in each mode or sub-model to determine for each driver which variables are useful for expressing the motion dynamics of that mode/sub-model. The variable selection result and the corresponding identified parameters are shown in Table 4 for all the drivers where the 0 (zero) parameter value means that the variable was not selected. Examining the Table 4, the range $u_1$ and time headway $u_6$ were selected in all modes/sub-models for all the drivers. This implies that the range ($u_1$) and time headway ($u_6$) are the most influential variables for expressing the motion dynamics in the car-following driving task. The jerk, $u_4$, was rarely selected across all modes amongst all drivers and, as such, is the variable with the least effect when expressing the dynamical motion characteristics of the car-following driving task.

In interpreting the identified parameters of a regression model, a positive value signifies a linear proportional relationship with the output, i.e., an increase in that input variable results in some increase in the output. On the other hand, an input variable with a negatively valued parameter represents an inverse relationship with the output variable, i.e., an increase in that input variable will lead to a decrease in the output variable. Examining the values of the identified parameters in Table 4, the range $u_1$ is positive in all the modes, meaning the speed increases as the range increases. However, the time headway $u_6$ has negative-valued parameters, and as such the speed decreases when the time headway increases. At first glance, this might seem counterintuitive as the speed should be expected to increase when the time headway increases. However, a careful consideration shows that the time headway acts as a suppressor that brings the speed back to its maximum limit. This idea is intuitive considering that the driving experiment was performed in the downtown area and the driving speed is limited to a maximum of about 40 km/hr. The parameters of the other variables can be interpreted in the same manner where an increase in positively valued parameters causes the output (speed) to increase while an increase in the negative value parameters causes the output to decrease.

Further observation shows that the combination of selected explanatory variables are different from mode (sub-model) to mode (sub-model) amongst all the drivers and are also different from one driver to another. In addition, both the signs and magnitudes of the parameters are different from one mode to the other, signifying that even the contribution or influence of a variable to the output can change from one sub-model to the other. The difference of results of the variable selection (i.e., which variables influence the output speed) from driver to driver agrees with the hypothesis of this paper about the need

for personalized automated and assistance systems, thus highlighting the importance of this work.

**Table 4.** Variable selection results with the identified parameters.

| Driver | $opt_s$ | Mode ($i$) | $u_1$ | $u_2$ | $u_3$ | $u_4$ | $u_5$ | $u_6$ |
|--------|---------|------------|-------|-------|-------|-------|-------|-------|
| A | 3 | 1 | 1.4078 | −0.1736 | −0.0468 | 0 | −0.1989 | −3.3325 |
|   |   | 2 | 0.4244 | 0 | 0.9857 | 0 | −0.1722 | −0.7046 |
|   |   | 3 | 1.2801 | −0.0813 | 0 | 0 | 0 | −2.8269 |
| B | 3 | 1 | 1.1372 | −0.3606 | 0.0466 | 0 | −0.5636 | −3.6705 |
|   |   | 2 | 0.7422 | 0 | 0 | −0.0442 | 0 | −0.5291 |
|   |   | 3 | 0.8931 | 0.2098 | 0.0122 | 0.1740 | 0.2186 | −2.7398 |
| C | 4 | 1 | 1.0559 | 0.1846 | −0.1030 | −0.2635 | 0.2396 | −3.1566 |
|   |   | 2 | 0.2212 | 0.1369 | 0.4913 | −0.0262 | 0 | −0.4392 |
|   |   | 3 | 0.7282 | 0.0547 | 0.0798 | 0 | 0 | −1.5486 |
|   |   | 4 | 1.0575 | 0.2268 | 0 | 0 | 0.3105 | −3.6367 |
| D | 3 | 1 | 1.0527 | −0.2552 | 0 | 0 | −0.0445 | −0.6431 |
|   |   | 2 | 1.2829 | 0.1571 | 0 | 0 | 0 | −3.5322 |
|   |   | 3 | 0.9368 | 0.1388 | 0.0282 | 0 | 0.2396 | −5.7718 |
| E | 3 | 1 | 1.3472 | 0 | 0 | 0 | −0.0929 | −5.2117 |
|   |   | 2 | 0.7686 | −0.2376 | 0.4456 | −0.0301 | −0.1950 | −0.9302 |
|   |   | 3 | 0.9445 | 0 | 0 | 0 | 0 | −2.9450 |
| F | 3 | 1 | 0.9255 | −0.3834 | 0.0337 | 0 | −0.6098 | −1.5482 |
|   |   | 2 | 0.9039 | −0.1619 | 0.1191 | 0 | 0 | −1.0215 |
|   |   | 3 | 0.9646 | −0.1796 | 0 | −0.3678 | 0 | −1.8258 |

## 5. Model Evaluation

### 5.1. Prediction Performance

The proposed model is first evaluated for modeling accuracy using prediction performance against the Gipps driver model [55]. This prediction performance was carried out on a data set different from the training data set. The Gipps driver model is chosen for comparison here because it is one of the most-used microscopic driver model for traffic flow simulation and the model output, speed is the same as that of the proposed model. Intuitively speaking, the Gipps model assumes that the ego car's driver selects his speed to ensure that the car can be safely brought to a stop if the leading vehicle suddenly comes to a stop. The Gipps car-following driver model is given as follows:

$$v_n(t + \tau) = \min(v_{following}, v_{free}),  \tag{13}$$

$$v_{following} = v_n(t) + 2.5 a_n \tau \left(1 - \frac{v_n(t)}{V_n}\right) \sqrt{0.025 + \frac{v_n(t)}{V_n}},  \tag{14}$$

$$v_{free} = b_n \tau + \sqrt{b_n 2 \tau^2 - \sqrt{b_n \left(2\left(x_{n-1}(t) - s_{n-1} - x_n(t)\right) - v_n(t)\tau - c\right)}}  \tag{15}$$

where $c = v_{n-1}^2(t)/\hat{b}$, and the variables in (14) and (15) are defined as in [55].

The result of prediction performance comparison is plotted in Figure 9 for Drivers B and C. The root mean square error (*rmse*), which measures the deviation of the predicted values from the actual values, is shown in Table 5 for all the drivers and is given as

$$rmse = \sqrt{\frac{\sum_{i=1}^{N}(\hat{y}_k - y_k)^2}{N}} \qquad (16)$$

where $\hat{y}_k$ and $y_k$ are the predicted and actual values, respectively, for the $k$th observation. Generally speaking, the lower the *rmse*, the better the model and, as can be seen from Table 5, the *rmse* of the proposed model is significantly lower compared to the Gipps model for all the drivers and across all modes. Additionally, in the proposed model, the *rmse* of the dangerous region (mode 1) is relatively lower compared to the other modes. Therefore, it can be said that the proposed model shows a better modeling accuracy than the Gipps driver model in modeling real-road driving.

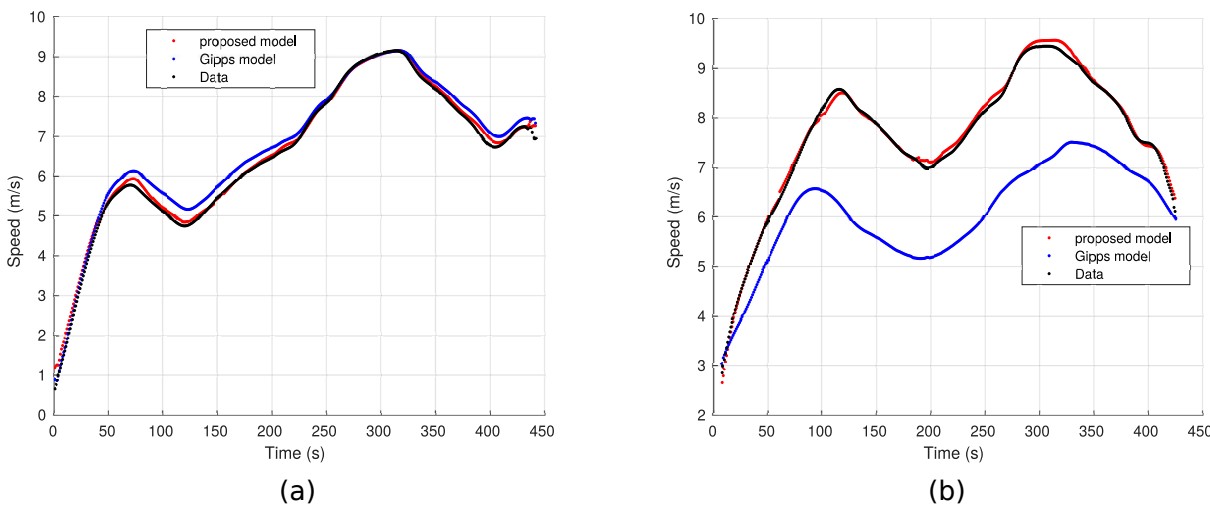

(a)         (b)

**Figure 9.** Prediction of real world test data by the proposed model and Gipps model: (**a**) using the test data from driver B's driving data; (**b**) using the test data from driver C's driving data.

**Table 5.** Root mean square error (*rmse*) prediction performance.

| Driver | Mode (*i*) | Gipps Model | Proposed Model |
|--------|-----------|-------------|----------------|
| A | 1 | 6.8086 | 0.0426 |
|   | 2 | - | - |
|   | 3 | 6.1663 | 0.3003 |
| B | 1 | 0.4176 | 0.1144 |
|   | 2 | 0.1421 | 0.2902 |
|   | 3 | 0.2351 | 0.0796 |
| C | 1 | 1.7185 | 0.0711 |
|   | 2 | - | - |
|   | 3 | 0.6308 | 0.0713 |
|   | 4 | 1.8837 | 0.1043 |
| D | 1 | 1.0527 | 0.1129 |
|   | 2 | 1.2829 | 0.1011 |
|   | 3 | 0.9368 | 0.1187 |
| E | 1 | 5.34527 | 0.0528 |
|   | 2 | - | - |
|   | 3 | 5.0658 | 0.3154 |
| F | 1 | 4.0194 | 0.1489 |
|   | 2 | 3.5276 | 0.1669 |
|   | 3 | 4.9792 | 0.1377 |

*5.2. Car-Following Simulation*

The applicability of the proposed model is investigated by closed-loop behavior verification. To this end, a car-following simulation was designed and implemented using both the identified (proposed) driver model and the Gipps driver model. To make the simulation as close to real-world autonomous driving as possible, the CARLA driving simulator in combination with a ROS (Robot Operating System) was adopted. Using the CARLA driving simulator, sensors such as 3D Lidar, Camera, GNSS can be mounted on the vehicles, and the data received from these sensors can be processed (mapping, localization, detection, etc.) using ROS. In addition, the vehicles used in the CARLA simulator have dynamics modeled from actual real-road vehicles. Thus, the framework used here for the simulation is as close as possible to that used in a real-world autonomous car-following task. The setup for the car-following simulation is shown in Figure 10.

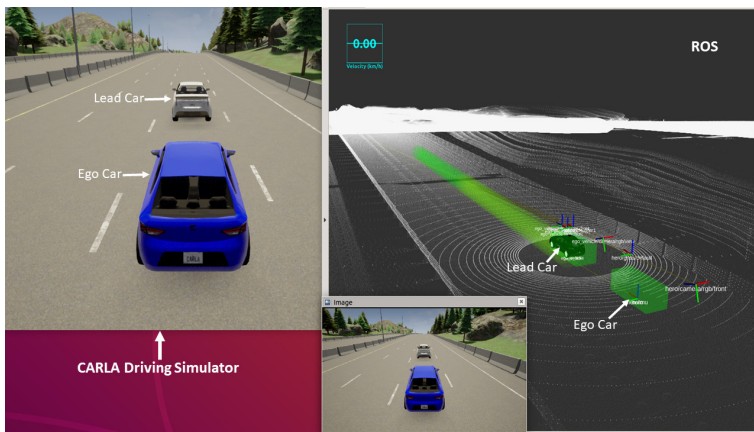

**Figure 10.** Setup used for the car-following simulation: A bridge between CARLA driving simulator and ROS.

This verification involves embedding together the identified driver model with the vehicle model and calculating the closed-loop behavior. The procedure for the verification is as follows:

**Step 1:** Initialize the states of the ego car and leading car, in this case, position and velocity.

**Step 2:** Based on the states, the input data of the driver model ($u_1 \sim u_6$) are computed.

**Step 3:** The mode is decided based on SVM and the corresponding input data of the driver model are updated.

**Step 4:** Using the identified driver model, the output (speed of ego car) is computed.

**Step 5:** Update the states of the ego and leading cars based on the output of Step 3 and the leading car's velocity pattern.

**Step 6:** Go to Step 2.

A similar procedure is adapted for the Gipps driver model by using the Gipps driver model to compute the output in Step 4, and removing Step 3.

The profiles from the simulation by using the identified driver model and the Gipps driver model together with the corresponding profiles from the real-road driving experiment data are shown Figure 11. It could be verified that the identified driver model was able to follow the leading car without collision, nor lagging far behind the leading car. However, even though the Gipps model could follow the leading car without any collision, although at some point it started lagging far behind the leading car. Furthermore, comparing the profiles to that of the actual real-road driving (Figure 11), the profiles of the identified model mimic better than that of the Gipps model. Therefore, it is confirmed that the original behavior is well reproduced by the identified model. A short video of the simulation is shown, respectively, https://youtu.be/jfyM5msDu2k (accessed on 18 May

2021) for Gipps' driver model and https://youtu.be/8Q0n3iaNbxE (accessed on 18 May 2021) for the proposed model.

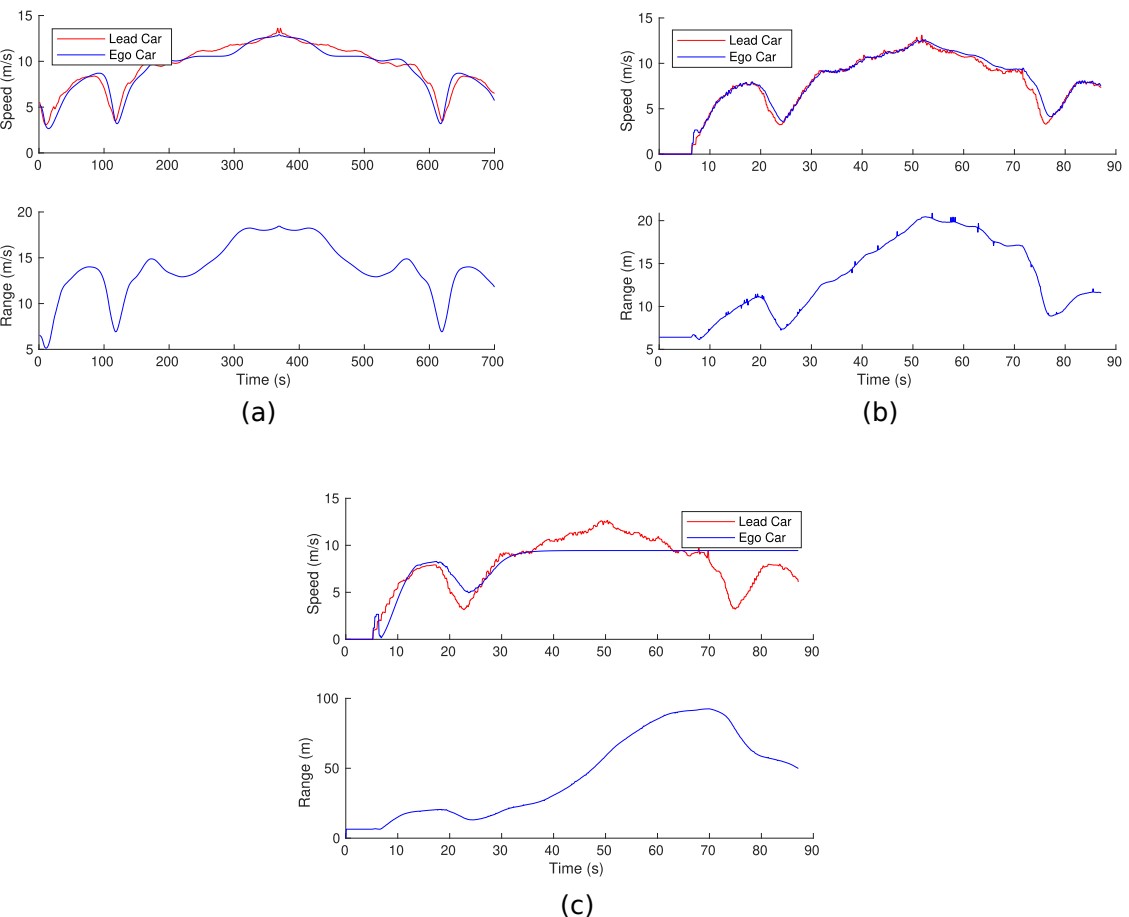

**Figure 11.** Profiles from the car-following simulation: (**a**) speed and range profiles from the actual real-road driving data; (**b**) speed and range profiles from the car-following simulation based on the proposed model; (**c**) Speed and range profiles from the car-following simulation based on Gipps' driver model.

## 6. Discussion

The useful applications of the proposed model are summarized as follows:

- The method presented in this study which is a combination of underlying dynamics identification and statistical model selection is well suited for understanding complex human driving behavior. The methodology proposed here was able to identify the variables responsible for the decision-making of the drivers in the car-following driving task. These variables as discussed in Section 4.1 are KdB ($u_3$) and range rate ($u_2$). In other words, the values of these variables decide how the motion dynamics are controlled by the driver or what variables to use for the vehicle's motion control. In addition, the variables range ($u_1$) and time headway ($u_6$) were identified using variable selection as the most influential variables in expressing the motion dynamics of the vehicle-following tasks for all the drivers. Furthermore, the model was able to identify the similarities (i.e., the decision-making variables and the influential motion dynamics variables) among the drivers, as well as the differences (i.e., the magnitudes of the decision-making variables and the differences in variable selection results) between drivers. Therefore, the direct application of this study is in designing controllers that can reflect drivers' preferences and characteristics for automated driving. For example, looking at Table 4, to design the control dynamics of the cruising region (mode 3) for driver B, the variable jerk ($u_4$) must be paid attention

to, whereas it is not necessary to consider this for driver A. These preferences in form of variable differences and the magnitudes of the parameters can be used for a personalized approach towards designing automated systems and ADAS. In addition, this method can be applied in order to gain knowledge about how the automation system/ADAS cooperate considering the human driver. Furthermore, by introducing flexibility in form of variable selection (i.e., using only the most influential variables) in the model, computational cost can be reduced, thereby facilitating online applications.

- This study presented a novel method for deciding the optimal number of behavioral segments (modes or clusters) from data based on consistent variable selection, thereby making it suitable for application in clustering or segmenting problems, particularly, where structural consistency of the identified clusters is of interest.

- Compared to the Gipps driver model, the proposed methodology was better both in prediction performance and in the car-following simulation, thereby, making it very suitable as a microscopic driver model in traffic flow applications, especially where expressing the actual driver behavior or personalized behavior generation is of importance.

- In the design of advanced driving assistance systems (ADAS) for car-following driving task, by using the work presented here, the driving situation (i.e., the mode) such as dangerous region, safe region, or cruising region could be identified either by using the specific explanatory variable's value or by looking at the values of the decision variables (i.e., KdB and range rate). The assistance system can be tuned appropriately based on the identified driving situation, hence enabling a better and more reliable system that is able to complement the human driver efficiently.

## 7. Conclusions

This paper proposed a novel methodology wherein the optimal number of sub-models of a PWARX is identified based on consistent variable selection. By leveraging the idea of consistent variable selection, an optimal model which is replicative and robust to a slight change of driving data can be realized. The PWARX model is well suited to describe both the decision-making and motion control facets in analyzing human driving behavior. The human driving behavior generally changes depending on the surrounding environment, driving time due to fatigue, or driver's mood, by using consistent variable selection with the PWARX in modeling a driver's driving behavior, a basal model that is robust to these changes, which in fact can be said to be the true model of that driver, can be identified.

The proposed model was applied to the car-following driving task resulting in the identification of three regions/modes/sub-models (dangerous, cruising, and safe). The risk feeling index, KdB $u_3$, and range rate $u_2$ Table 4 were found to play important roles in the decision-making to switch the control laws between the motion facets. By using statistical variable selection, the range $u_1$ and time headway $u_6$ Table 4 were found to be the most influential variables in the motion control of the car-following task. In addition, the similarities in form of identifying the decision-making variables and the influential motion dynamics variables among drivers and the differences between them by looking at the magnitudes of the decision-making variables and the differences in variable selection results could be identified.

Furthermore, the identified model for the car-following driving task was compared to the conventional microscopic driver model, the Gipps driver model, both in real-road driving data prediction and CARLA-ROS based car-following simulation. The comparison shows that the proposed model shows significantly better prediction performance and is able to mimic the real-road driving behavior better when simulated.

The content of this paper can be summarised as follows. Section 1 introduced the background, motivation, and the related works of this study. The proposed method of this study was introduced and discussed in Section 2, and Section 3 applied the proposed model to the car-following driving task. Finally, Section 6 discussed some of the applications of the proposed method in this paper.

**Author Contributions:** Conceptualization, all authors; methodology, J.C.N.; formal analysis, J.C.N.; research, all authors; resources, H.O. and T.S.; data curation, J.C.N.; writing—original draft preparation, J.C.N.; writing—review and editing, all authors; supervision, H.O. and T.S. All authors have read and agreed to the published version of the manuscript.

**Funding:** This research received no external funding.

**Institutional Review Board Statement:** The study was conducted according to the guidelines of the Declaration of Helsinki, and approved by the Ethical Committee of Graduate School of Engineering, Nagoya University with approval number of 18-6.

**Informed Consent Statement:** Informed consent was obtained from all subjects involved in the study.

**Data Availability Statement:** The data presented in this study are available on request from the corresponding author. The data are not publicly available due to privacy restrictions.

**Conflicts of Interest:** The authors declare no conflict of interest.

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
