# Peer review of "Driving Behavior Modeling Based on Consistent Variable Selection in a PWARX Model"

_applsci, doi:10.3390/app11114938_

Round 1

Reviewer 1 Report

The article focuses mainly on the model, supporting it with experimental research. However, the results obtained from the experiments were treated superficially. The issues related to both the vehicle motion parameters and the driver's behavior have not been developed either.

In the introduction, no review of the current achievements in the field of:

- autonomous vehicles / the authors refer to it in line 18, currently vehicles are approved for use in selected locations in the USA /,

- LIDAR technology,

- experimental tests in the field of vehicle traffic in road conditions / in the city, on the motorway, on various surfaces, e.g. snow, ice, other pollutants /,

- experimental tests related to the strategy of behavior of the driver in conditions of reduced adhesion / e.g. Driver's strategy and braking distance in Winter /,

- traffic simulators,

- analysis of the driver's behavior on the road, - the impact of the road situation on vehicle movement and driver behavior,

- analysis of the mental state of the driver conditioned by driving time, road situation, random factors / e.g. other vehicles, cyclists, pedestrians, animals /, which affects the driving style / e.g. aggressive style /,

- analysis of the driver's reaction time depending on road conditions.

There is also a lack of a full introduction to simulation methods and methods of assessing driver behavior on the road. It is also unknown what parameters will be taken into account in the model and why. The presented description is largely based on the literature from before 2016. This results in a lack of reference to current traffic models and analyzes of driver behavior.

Chapter 2 lacks a clear demonstration of the adopted boundary conditions and assumptions for the model. The description of the model is superficial and hardly readable for the recipient. No information is available about what input data will be taken into account in the model, what weights / priorities are. What factors must be included in the model? Which can be omitted due to the lack of influence on the result? What simplifications have been used? How was the driver's behavior interpreted mathematically?

Chapter 3 lacks information about the road that the test drivers were to travel on, about its nature, about the road situations encountered. How many times has the test vehicle stopped / high beam, traffic situation /? What was the topography? What was the research section like / a map with the route would be useful /? How was driver behavior data collected? How did the driver's behavior affect the vehicle's movement? How was driving style / driver type identified? What traffic parameters were collected and taken into account in the model?

The presented figures 4, 5 and 6 are hardly understandable. All values ​​in Tables 2 and 3 are converted to model values ​​and cannot be related to motion parameters. Only Figure 7 makes it possible to assess the correlation between the experimental data and the proposed model. Chapter 4 is hardly readable for the recipient.

Chapter 5 compares the model with the Gipps model. Such a comparison can only indicate whether the developed model is better, worse or at a similar level. The most important thing in this case is the comparison with the actual research. They reflect the real parameters of the vehicle motion and the driver's behavior on the road. What is the car following simulation for? Is there a follow-up control in mind - tracking the selected vehicle? Such tracking systems are already installed in vehicles, as well as systems for keeping the car in the lane, assisting the braking process and assisting overtaking and parking. Inability to watch the test video without logging in.

Chapter 6 does not show that the proposed model is well suited to understanding complex driver behavior. In the previous chapters there is no reference to the driver's behavior. Everything is based on the vehicle's motion and tracking parameters. The driver's behavior and psychophysical state were only reduced to the traffic parameters. The state in which the proposed model can identify the variables responsible for drivers' decision-making is also not indicated. How were similarities and differences between the drivers identified? How can driver preferences and characteristics for automated driving be reflected? Too little was discussed the innovative aspects of the presented model. What's the concrete solution? What's new? There is no extensive comparative analysis with other models and literature.

The whole of Chapter 7 needs to be rebuilt. The current content does not reflect the content of the article. After introducing the necessary changes, it will be possible to refer to the driver's strategy, as well as the obtained traffic parameters in given traffic conditions, e.g. city traffic.

Literature

Autocit 4/30

Acceptable number of self-citations.Ideally, it would be below 10%.

26/30 from before 2016.

10/30 from before 2000.

There is no current literature on the issue.

Authors should try to ensure that the literature older than 5 years does not exceed 25%, currently it is almost 87%.

Reviewer 2 Report

In this work a hybrid human driving model is proposed. The approach focuses on automated mode segmentation and optimal selection of the number of modes.
Although the paper is well structured, some issues must be addressed, especially the presentation and discussion of the results what is a key aspect. They are listed below:

- Please expand the acronyms the first time they appear in the text. PWARX and ARX are firstly appear in abstract without description.
- The ideas introduced in section I are clear. However, the last paragraph of this section (except the last sentence) just repeat what is already writen above. Please rewrite the last paragraph to increase the readability of the paper.
- In pp4, line 122, n = pny + mnu, please specify what n, p and m are.
- The optimal number of sub-models selection is one of the claims of the paper. However, in section 2.4 where this decision strategy is introduced, Algorithm 1 not explained in the text at all. This must be corrected.
- Notation used in section 3.2 share variable names (x, u, etc.) with previous definitions that seem to refer to completely different magnituds. Please use other variable names or at least specify that in section 3.2 variables name refer to other definition explicitly.
- In section 4, "drivers A-F" are mentioned but not described before. The same happens to the different "modes" applied. This makes imposible to undestand the text. What is the difference among the considered drivers? Why 6 drives? What is the difference among the considered modes?
Figures 4-6 and Tables 2-3 are supposed to represent a big part of the results of this paper however they are barely described in the text. In fact nothing about the values shown in tables is dicussed in this section. This section should be rewriten/reordered to make it understandable.
- In section 5.2, it is said that "the profiles of the identified model agrees well.". Well in what sense? Statements such as these show a clear lack of scientific rigour when analysing the results obtained. 
- The videos linked in section 6, are not accessible.

Typos:
- pp. 3, line 63: "The results shown is ..." -> are
- pp. 4, line 138: "neighboring c data" what is "c"?
- pp. 14, line 296: "It could be the verified that"

Reviewer 3 Report

The paper proposes a hybrid method for modelling driver behavior. The approach is interesting, but I proposes the following changes to be implemented prior to publication.

- Switching mechanisms to represent driver behavior have already been introduced (below) (that work is also based on real driving data).
- I like the fact that this research uses an open source approach with CARLA.

- The paper uses a wide variety of techniques. There should be more comments giving some narrative to the scope of the paper; so that it is more clearly understood why these particular technique selection is employed.
- The paper should discuss more widely potential limitations of the approach, so that these can be compared to other methods.
- How where all the hyperparameters of the data pipeline selected and tuned?
- Related references are missing, such as the hybrid driver behavior models proposed in 'Modeling lane keeping by a hybrid open-closed-loop pulse control scheme' and novel steering systems that can better accommodate ADAS: 'Communication and interaction with semiautonomous ground vehicles by force control steering'.

Round 2

Reviewer 1 Report

Most of the authors took into account the comments from the reviews. The amount of new literature cited has improved, however, the balance of literature older than 5 years is still 50%.

Reviewer 2 Report

In this work a hybrid human driving model is proposed. The approach focuses on automated mode segmentation and optimal selection of the number of modes. In this revised version of the manuscript authors have addressed my previous concerns, making the paper more complete and understandable. Neverthess, before publication, I suggest the authors to include some of the recent advances on decision-making architectures for automated driving (such as https://ieeexplore.ieee.org/abstract/document/8814070) in the introduction to provide a better contextualization of the proposed approach.

Reviewer 3 Report

I think the authors have improved the manuscript and answered the reviewer's questions. Hence I recommend the paper for publication.

Author Response

No modifications were requested by the reviewer, therefore, no response required. 
Thank you.